# Quantification of Ceftaroline in Human Plasma Using High-Performance Liquid Chromatography with Ultraviolet Detection: Application to Pharmacokinetic Studies

**DOI:** 10.3390/pharmaceutics13070959

**Published:** 2021-06-25

**Authors:** Ana Alarcia-Lacalle, Helena Barrasa, Javier Maynar, Andrés Canut-Blasco, Carmen Gómez-González, María Ángeles Solinís, Arantxazu Isla, Alicia Rodríguez-Gascón

**Affiliations:** 1Pharmacokinetic, Nanotechnology and Gene Therapy Group (Pharma Nano Gene), Centro de Investigación Lascaray Ikergunea, Faculty of Pharmacy, University of the Basque Country UPV/EHU, 01006 Vitoria-Gasteiz, Spain; ana.alarcia@ehu.eus (A.A.-L.); marian.solinis@ehu.eus (M.Á.S.); arantxa.isla@ehu.eus (A.I.); 2Instituto de Investigación Sanitaria Bioaraba, 01009 Vitoria-Gasteiz, Spain; helena.barrasagonzalez@osakidetza.eus (H.B.); franciscojavier.maynarmoliner@osakidtza.eus (J.M.); andres.canutblasco@osakidetza.eus (A.C.-B.); carmen.gomezgonzalez@osakidetza.eus (C.G.-G.); 3Intensive Care Unit, Araba University Hospital, Osakidetza Basque Health Service, 01009 Vitoria-Gasteiz, Spain; 4Microbiology Service, Araba University Hospital, Osakidetza Basque Health Service, 01009 Vitoria-Gasteiz, Spain

**Keywords:** ceftaroline fosamil, HPLC, pharmacokinetics, pharmacokinetics/pharmacodynamics

## Abstract

This study was conducted to develop a rapid, simple and reproducible method for the quantification of ceftaroline in plasma samples by high-performance liquid chromatography with ultraviolet detection (HPLC-UV). Sample processing consisted of methanol precipitation and then, after centrifugation, the supernatant was injected into the HPLC system, working in isocratic mode. Ceftaroline was detected at 238 nm at a short acquisition time (less than 5 min). The calibration curve was linear over the concentration range from 0.25 to 40 µg/mL, and the method appeared to be selective, precise and accurate. Ceftaroline in plasma samples was stable at −80 °C for at least 3 months. The method was successfully applied to characterize the pharmacokinetic profile of ceftaroline in two critically ill patients and to evaluate whether the pharmacokinetic/pharmacodynamic (PK/PD) target was reached or not with the dose regimen administered.

## 1. Introduction

Ceftaroline is a broad-spectrum fifth generation cephalosporin with bactericidal activity against Gram-positive organisms, including methicillin-resistant *Staphylococcus aureus* (MRSA) and drug-resistant *Streptococcus pneumoniae*, as well as non-beta-lactamase-producing Gram-negative organisms [1,2,3]. Ceftaroline acts to inhibit the growth of bacterial cells by interfering in the synthesis of the cell wall. It has a high affinity for modified penicillin-binding proteins (PBPs), such as PBP2a in *S. aureus* and PBP2x in *S. pneumoniae*, leading to high activity against resistant Gram-positive cocci [4,5,6].

This new antibiotic was approved in 2010 by the Food and Drug Administration (FDA) and in 2012 by the European Medicines Agency (EMA). The indications are community-acquired bacterial pneumonia (CABP) and acute bacterial skin and skin structure infections (ABSSSI) [7,8]. A new label expansion to include bacteremia by *S. aureus* was approved in 2015 by the FDA. More recently, ceftaroline fosamil was also approved for pediatric populations (≥2 months to <18 years old) by both the FDA and EMA [9,10].

Ceftaroline fosamil, the prodrug of the active metabolite ceftaroline, is quickly hydrolyzed by plasma phosphatases. Additionally, an inactive metabolite is generated by the hydrolysis of the beta-lactam ring (Figure 1) [11,12,13]. Ceftaroline is administered at a dose of 600 mg every 8 or 12 h as a 1 h intravenous infusion. The duration of the treatment is 5–14 days for ABSSSI and 5–7 days for CABP. The main elimination route is through the kidney, and dose adjustment is required if the patient presents moderate or severe renal insufficiency [12]. Ceftaroline clearance is significantly affected by age, renal function or the presence of infection [14]. As a beta-lactam antimicrobial, the pharmacokinetic/pharmacodynamic (PK/PD) index that better correlates with efficacy is the time, expressed as the percentage of the dosing interval by which free drug concentration is over the minimum inhibitory concentration (MIC) of the bacteria responsible for the infection (%*f*T_>MIC_ ) [15,16].

Table 1 features some of the physicochemical properties of ceftaroline that help the development of analytical procedures. Literature about analytical methods to quantify ceftaroline in biological samples is very scarce. A recent study [17] described a method based on on-line solid phase extraction coupled with high-performance liquid chromatography-tandem mass spectrometry (HPLC-MS/MS); however, this methodology requires expensive equipment and skilled operators, resulting in recurring equipment and staff costs; therefore, it is necessary to design and validate new methods that are more affordable and easier to use. 

Here we propose a simple and rapid method based on HPLC with ultraviolet (UV) detection to quantify ceftaroline in plasma, which can be easily used to maximize the follow-up of infected patients by way of therapeutic drug monitoring. The method was validated according to the FDA and EMA guidelines [21,22] and applied to the characterization of the pharmacokinetic profile of ceftaroline in two critically ill patients; that is, patients who presented one or more organic dysfunctions, and who may have presented alterations in the pharmacokinetics (PK) of the drugs used, including antibiotics, due to the presence of these dysfunctions and/or due to the treatments they required for their pathologies.

## 2. Materials and Methods

### 2.1. Chemicals, Reagents and Samples

Ceftaroline dihydrochloride was kindly supplied by Pfizer Inc. Acetonitrile HPLC gradient (ACN) and methanol were purchased from Scharlau (Barcelona, Spain), and ammonium dihydrogen phosphate was purchased from Sigma-Aldrich Chemie Gmbh (Steinheim, Germany). The ultrapure water was obtained from the Mili-Q^®^ Plus apparatus (Millipore, Burlington, MA, USA). 

Blank plasma from healthy donors, used to prepare calibration standard and quality control (QC) samples, was provided by the Basque Biobank (www.biobancovasco.org, accessed on 21 May 2021) and was processed following standard operation procedures with appropriate approval from the Ethical and Scientific Committees (Code CES-BIOEF 2020-34). 

Plasma from critically ill patients was processed following a protocol previously approved by the Basque Clinical Research Ethics Committee (EPA2018019 (SP)). Samples and data from these patients were provided by the Basque Biobank (www.biobancovasco.org, accessed on 21 May 2021) and were processed following standard operation procedures with appropriate ethical approval.

### 2.2. Chromatographic Equipment and Conditions

The chromatographic system used was a Waters^TM^ 1525 binary HPLC pump connected to an in-line degasser, an autosampler (2707) and UV/visible detector (2498). The HPLC system was controlled with the Waters Breeze HPLC software (Version 6.20.00.00., Waters, Milford, MA, USA), which was also used to process the data. The final conditions of the method were selected from preliminary experiments, in which different columns, mobile phase composition, flow rate and sample preparation conditions were tested. A UV spectrum was used in order to select the wavelength for detection. Chromatographic analysis was performed using a Symmetry C18 (5 µm × 4.6 mm × 150 mm) column (Waters). The mobile phase for ceftaroline determination consisted of ammonium dihydrogen phosphate buffer:acetonitrile (85:15, *v:v*). Buffered solution was prepared by dissolving 575 mg of ammonium dihydrogen phosphate in 1000 mL of ultrapure water. Later it was filtered and degassed in an Ultrasons ultrasonic bath (Selecta, Barcelona, Spain) and delivered with a flow rate of 1 mL/min. The assay was performed at room temperature (RT) and the selected wavelength to detect ceftaroline was 238 nm.

### 2.3. Working Solutions, Calibration Curves and Quality Control Samples

A ceftaroline stock solution was prepared every day by dissolving ceftaroline dihydrochloride in a mixture of water:methanol (1:1; *v:v*) to obtain a concentration of 1 mg/mL. For the standard calibration samples, the stock solution was diluted with water to prepare the working solutions (400, 200, 100, 50, 10, 5 and 2.5 µg/mL). A total of 100 µL of every working solution was mixed with 900 µL of drug-free human plasma to obtain the standard calibration samples (40, 20, 10, 5, 1, 0.5 and 0.25 µg/mL). 

Three QC samples (high, medium and low) were prepared. The stock solution was diluted with water to prepare the working solutions (300, 150 and 7.5 µg/mL), and 100 µL of every working solution was mixed with 900 µL of drug-free human plasma to obtain the QC samples (30, 15 and 0.75 µg/mL). 

### 2.4. Sample Preparation

A 100 µL aliquot of the ceftaroline plasma samples was mixed with 200 µL of methanol and centrifuged for 10 min at 10,000× *g* at 4 °C. The supernatant was collected and a volume of 20 µL was injected onto the HPLC. 

### 2.5. Method Validation

The assay was validated according to the regulatory guidelines on bioanalytical method validation of the FDA (2018) and the EMA (2012) [21,22].

Complete calibration curves over the concentration range of 0.25–40 µg/mL were analyzed on three correlative days. A linear regression with a weighting factor of 1/concentration was used to plot the peak area of ceftaroline (response) versus the corresponding concentration. Slopes, y-intercepts, correlation coefficient (R), the relative error (RE, %) from the nominal level of each standard and the coefficient of variation (CV, %) of the response factors (chromatographic area/concentration) were calculated. The correlation coefficient (R) had to be greater than or equal to 0.99 and the RE within 15%, except for the lower limit of quantification (LLOQ). The CV of the response factor should was intended to be within 15%.

Selectivity was determined using plasma samples of six healthy individuals. Additionally, plasma samples from six critically ill patients not receiving ceftaroline were used. Specificity was evaluated by testing both matrices regarding interference near the retention time of ceftaroline under the used chromatographic conditions.

The precision and accuracy of the method were determined using QC samples. Over 3 days, five QC samples at the three concentration levels (0.75, 15 and 30 μg/mL) were analyzed. Intra- and inter-day precision were calculated as the coefficient of variation (CV, %) within a single run and between the three assays, respectively; for the calculation of intra- and inter-assay accuracy, the RE of the nominal concentration was calculated. Analytical series were considered approved if the RE and CV did not exceed ±15%.

As with the QC samples, the intra- and inter-assay precision and accuracy of the LLOQ were also calculated. The LLOQ was considered the lowest level included in the calibration curves. The RE and CV were intended not to exceed ±20%.

### 2.6. Stability

For stability studies, QC samples were prepared at the same concentration levels as the QC used in the accuracy and precision study. The stability of ceftaroline in plasma samples was evaluated after storage at −20 °C and −80 °C. Stability was also determined after three freeze–thaw cycles. Three samples from each QC were subjected to three freeze–thaw cycles and analyzed after the third cycle. Samples were thawed at room temperature. 

Post-preparative stability, that is, the stability of ceftaroline in the processed samples, was evaluated by maintaining it immediately after preparation at 4 °C. 

The stability of ceftaroline during sample processing was studied as well. QC samples were kept at room temperature for 2 h, then processed and analyzed using the chromatographic method. 

Ceftaroline in the samples was considered stable when the concentration at each level was within ±15% of the nominal concentration.

### 2.7. Application of the Method to Pharmacokinetic Studies

We evaluated the applicability of the method for PK studies by analyzing plasma samples collected from two critically ill patients diagnosed with pneumonia and treated with ceftaroline fosamil (600 mg every 8 h) after positive cultures of methicillin-resistant *Staphylococcus aureus* (MRSA). Blood samples were collected in K_2_EDTA tubes and centrifuged at 5000× *g* for 10 min. Plasma samples were stored at −80 °C until analysis and were analyzed within three weeks of extraction. Table 2 shows data from ceftaroline-treated patients. The study was conducted among critically ill patients admitted to the ICUs of Araba University Hospital (Vitoria-Gasteiz, Spain). Informed consent was obtained from both subjects involved in the study.

From the plasma concentrations of ceftaroline in the patients, we obtained the individual pharmacokinetic parameters through a non-compartmental analysis. For this purpose, we used the software Phoenix 64 (Build 8.1.3530, Certara USA, Inc., Princeton, NJ, USA). The elimination rate constant (Ke) was obtained by log-linear regression analysis of the terminal phase of the plasma drug concentration–time curve. The half-life (t_1/2_) was obtained by using the following equation: t_1/2_ = ln(2)/ke. The area under the plasma concentration–time curve from the first to the last concentration measured (AUCτ) was calculated with the linear trapezoidal method. The total body clearance (CL_T_) was obtained with the following equation: CL_T_ = dose/AUCτ. The mean residence time (MRT) was obtained through the equation: MRT = AUMCτ/AUCτ, where AUMCτ is the area under the moment curve. Finally, the distribution volume at steady-state (Vss) was estimated with the following equation: Vss = CL_T_ × MRT.

## 3. Results

The analytical HPLC-UV method was evaluated for selectivity, linearity, precision and accuracy. The stability of the stored samples, during sample processing and in the chromatographic system was also evaluated.

### 3.1. Chromatography and Detection

Figure 2 shows representative chromatograms of a blank sample from a healthy donor, a sample from a critically ill patient not receiving ceftaroline, the LLOQ (0.25 µg/mL), a calibration standard sample (20 µg/mL) and a sample of a critically ill patient treated with ceftaroline.

As the chromatograms show, ceftaroline presented a retention time of 4.66 ± 0.01 min. No interfering peaks were observed either in the samples obtained from healthy subjects or in the samples from critically ill patients not treated with ceftaroline.

### 3.2. Validation

Calibration curves were linear over the concentration range of 0.25–40 µg/mL. They were satisfactory fitted by linear regression with 1/the concentration weighting factor. Table 3 shows the parameters of the three calibration curves used for the linearity study. The coefficients of correlation (R) were always ≥0.99, and the CV of the response factor was <11%. Moreover, the RE of every standard was always <7%.

The precision and accuracy of the QC samples and the LLOQ for ceftaroline are given in Table 4. The intra-assay precision, expressed as the CV, was always <4%, and the inter-assay precision was <6%. Intra-day inaccuracy (RE, %) ranged from 1.35 to 3.52%, and inter-day inaccuracy ranged from 2.37 to 6.47%. The intra- and inter-assay precision levels for the LLOQ were 2.62 and 3.77%, and the intra- and inter-assay inaccuracies were 2.96 and 4.08%, respectively. Therefore, precision and accuracy were in accordance with the guideline acceptance criteria (≤15% for the QC samples and ≤20% for the LLOQ).

### 3.3. Stability

The results of stability assessment of ceftaroline under various conditions are illustrated in Table 5. Under storage, ceftaroline in plasma samples was stable for up to 2 weeks at −20 °C, and for at least 3 months at −80 °C. The stability in plasma after three freeze and thaw cycles and thawing at room temperature was checked and the variation was within 15% of the nominal concentration. Moreover, stability during sample processing was also confirmed. Stability in the autosampler was guaranteed at 4 °C during 8 h.

### 3.4. Analysis of Patient Samples

The described method was used to analyze plasma samples from two critically ill patients treated with ceftaroline fosamil (600 mg every 8 h). Each analytical batch included a blank sample, seven standard calibration samples, six QC samples (two of each concentration level), and plasma samples of the patient. Acceptance or rejection of the analytical batch was based on the coefficient of correlation (R > 0.99), and the RE of the standard calibration and the QC samples (≤15%). Figure 3 shows the concentration–time profile of ceftaroline in the two patients.

Table 6 shows the pharmacokinetic parameters of ceftaroline in the two critically ill patients obtained by non-compartmental analysis.

## 4. Discussion

In this paper, we describe a rapid, simple and reproducible method for the quantification of ceftaroline in plasma. This method is applicable to pharmacokinetic studies and it presents several advantages: (i) the processing of the samples is based on a simple protein precipitation, avoiding more complex and time-consuming steps, such as solid-phase extraction or liquid–liquid extraction; (ii) it works in isocratic mode; (iii) it has a short acquisition time (less than 5 min); and (iv) it includes ultraviolet detection. Reported methods for quantifying ceftaroline in biological samples are scarce, and they are based on mass spectrometry [17]. Liquid chromatography coupled to mass spectrometry (LC-MS/MS) is an increasingly important tool in therapeutic drug monitoring as it offers increased sensitivity and specificity compared to other methods. However, it presents important disadvantages, such as matrix effects, expensive equipment cost, time-consuming optimization requirements, and the necessity of well-trained personnel [23].

The selectivity of the method was demonstrated by the absence of interfering peaks in plasma from healthy donors and from critically ill patients not treated with ceftaroline. Ceftaroline is eliminated mainly by the renal route, and only a small fraction is converted into an inactive metabolite, which presents higher polarity; therefore, interference with ceftaroline is not expected. In spite of the use of ultraviolet detection, the LLOQ established in our method (0.25 µg/mL) was comparable to that reported by other authors applying a LC-MS/MS method (0.2 µg/mL) [17], and it was low enough to precisely and accurately quantify the minimum concentrations reported in pharmacokinetic studies, both in healthy subjects and patients [12,24]. Moreover, the LLOQ was adequate for detecting ceftaroline underexposure, since it was lower than the clinical breakpoint for *Staphylococcus aureus* (1 mg/L) and equal to the clinical breakpoint for *Streptococcus pneumoniae* reported by the European Committee of Antimicrobial Susceptibility Testing (EUCAST) [25]. *S. aureus* and *S. pneumoniae* are two of the main microorganisms responsible for the infection of patients for whom ceftaroline is indicated.

Calibration curves were linear over the concentration range from 0.25 to 40 µg/mL. The upper limit was adequate considering the plasma concentration expected in humans after intravenous administration of ceftaroline fosamil. In a previous study [12], ceftaroline C_max_ in healthy subjects was 10.0 ± 0.8, 19.0 ± 0.7 and 31.5 ± 2.4 µg/mL for treatment with 300 mg every 12 h, 600 mg every 12 h and 800 mg every 24 h, respectively. In the same study, C_max_ of ceftaroline in subjects with severe renal impairment (CrCL ≤ 30 mL/min) treated with 400 mg was 17.9 ± 2.9 µg/mL.

As our results show, the method is precise and accurate. Stability studies revealed that ceftaroline in plasma samples stored at −20 °C and at −80 °C was stable for up to 15 days and at least three months, respectively. Moreover, ceftaroline was stable after three freeze–thaw cycles, which confirms that it can be adequately quantified if a re-analysis of the sample is required.

Once the method was validated, it was applied to quantify ceftaroline in plasma samples from two critically ill patients diagnosed with pneumonia and treated with 600 mg of ceftaroline fosamil every 8 h. The samples were adequately analyzed, resulting in the concentrations of all samples being within the linearity range of the calibration curve. From the concentration data, a non-compartmental analysis was carried out to determine the pharmacokinetic parameters. The two patients presented a similar value of CL_T_ (8.84 and 7.33 L/h). The highest differences in the PK parameters of the two patients were detected for C_max_ (18.22 and 30.28 µg/mL) and for Vss (36.40 and 23.61 L). In this sense, it is important to take into account that pharmacokinetics of drugs in critically ill patients is highly variable [26,27,28,29,30]. The pharmacokinetic parameters of ceftaroline in our patients were of the same order as those reported in healthy subjects [12]. A recent study [24] described the PK of ceftaroline in critically ill patients undergoing continuous renal replacement therapy and treated with ceftaroline (400 mg every 8 h or 400 mg every 12 h). In spite of the different characteristics of these patients, the CL_T_ reported (ranging from 6.99 to 8.02 L/h, *n* = 4) was similar to that obtained in our patients (8.84 and 7.33 L/h). To the knowledge of the authors, no other study has reported the pharmacokinetics of ceftaroline in critically ill patients.

Ceftaroline exhibits time-dependent bacterial killing, and a successful outcome is associated with the percentage of time of the dosing interval in which the unbound serum antibiotic concentration remains above the minimum inhibitory concentration (%*f*T_>MIC_). For beta-lactams, the %*f*T_>MIC_ value needed for bacterial activity is between 40 and 70% in in vivo infection models, although clinical data suggest that optimal efficacy is achieved when %*f*T_>MIC_ is 100% [31]. Considering the through concentrations measured in the patients included in the study (2.34 and 2.87 mg/L, just before the following dose), the low protein binding [6], and the MIC value for MRSA (0.5 and 0.25 mg/L), the PK/PD target (%*f*T_>MIC_ of 100%) was met; therefore, we can conclude that they were adequately treated with ceftaroline. In fact, the dose administered (600 mg every 8 h) would cover for an MIC value ≤ 2 mg/L. In a recent study [9], ceftaroline demonstrated potent in vitro activity against a large collection of *S. aureus* isolates recovered worldwide, including methicillin-susceptible (MSSA) and methicillin-resistant (MRSA). That study reported MIC_90_ values for MRSA ranging from 1 to 2 mg/L.

Approximately 64% of the ceftaroline dose is excreted renally unchanged, and the CL_T_ is dependent on the CrCL of the patient [12]. Our patients presented high renal function and one of them, with CrCL of 129 mL/min/1.73 m^2^, was within the limit of augmented renal clearance (ARC). It is known that about 20–65% of critically ill patients present ARC, defined as the clinical situation in which CrCL is ≥130 mL/min/1.73 m^2^ [32]. This phenomenon may lead to subtherapeutical concentrations and worse clinical outcomes when following standard dosing guidelines. It is particularly important for antibacterial agents that are eliminated by the kidney and whose activity is time-dependent [26], as in the case of ceftaroline. Therefore, for patients with ARC, therapeutic monitoring of ceftaroline could be beneficial in preventing therapeutic failure [32]. The method we describe in this work to measure plasma concentrations may be very useful for monitoring in routine clinical practice in critically ill patients with ARC.

It is well-known that the variability of PK of drugs in critically ill patients is very high [33], and clinical trials to estimate the PK parameters in this population must include a high number of patients; additionally, population models are recommended. In this regard, our study presents a limitation. However, our purpose was to demonstrate the usefulness of the method better than a full characterization of the ceftaroline PK in critically ill patients.

In conclusion, we developed and validated a rapid and sensitive HPLC-UV method for quantification of ceftaroline in plasma samples. Simple sample preparation and short acquisition time enabled a high sample throughput while remaining cost-effective, which makes this method very useful for pharmacokinetic studies and therapeutic drug monitoring. The method was successfully applied to characterize the pharmacokinetic profile of ceftaroline in two critically ill patients and to evaluate whether the PK/PD target was reached or not with the dose regimen administered.

## Figures and Tables

**Figure 1 pharmaceutics-13-00959-f001:**
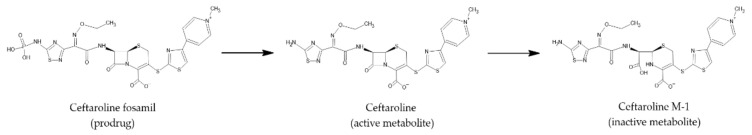
Chemical structure of ceftaroline fosamil, ceftaroline and ceftaroline M-1.

**Figure 2 pharmaceutics-13-00959-f002:**
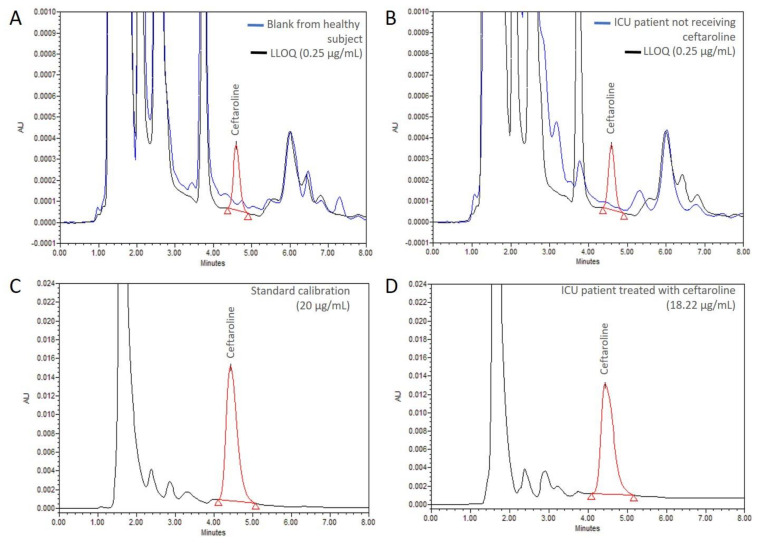
Representative chromatograms. (**A**) Blank sample from a healthy subject and the LLOQ. (**B**) Blank sample from an ICU patient not treated with ceftaroline and the LLOQ. (**C**) Standard calibration sample (20 µg/mL). (**D**) Sample of an ICU patient treated with ceftaroline (18.22 µg/mL). ICU: intensive care unit.

**Figure 3 pharmaceutics-13-00959-f003:**
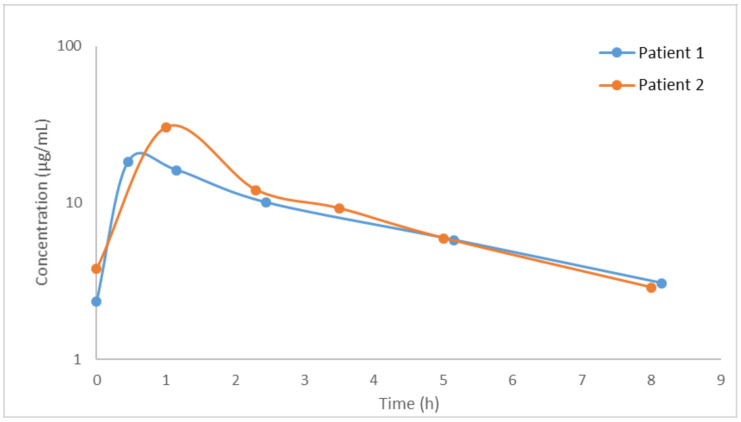
Plasma concentration of ceftaroline in the two critically ill patients.

**Table 1 pharmaceutics-13-00959-t001:** Physicochemical properties of ceftaroline.

Molecular Weight (g/mol) [18]	Solubility (mg/mL) [18]	pKa (Strongest Acidic) [19]	pKa (Strongest Basic) [19]	logP [19]	logP [20]
684.7	>100	1.31	0.31	−2.7	−0.84

**Table 2 pharmaceutics-13-00959-t002:** Data for the critically ill patients.

Patient	1	2
Age (year)	72	74
Weight (kg)	90	75
Sex	M	M
Body surface area (m^2^)	2.06	1.86
Dosage	600 mg every 8 h	600 mg every 8 h
Previous doses	4	3
APACHE II score	14	30
Creatinine clearance (mL/min/1.73 m^2^)	129	87
Glucose (mg/dL)	110	94
Albumin (g/dL)	2	2.7
Total protein (g/dL)	5.9	4.8
Haemoglobin (g/dL)	10.8	9.5
Haematocrit (%)	34.7	29.8
Ceftaroline MIC (MRSA) (mg/L)	0.5	0.25

APACHE: acute physiology and chronic health evaluation; MIC: minimum inhibitory concentration.

**Table 3 pharmaceutics-13-00959-t003:** Mean parameters of the calibration curves for ceftaroline in plasma.

Y = bx + a	R1	R2	R3
a	−681	87.9	1310
b	15,200	15,800	15,700
R	0.999	0.998	1.000
Response factor (CV, %)	6.72	5.71	10.78

**Table 4 pharmaceutics-13-00959-t004:** Intra- and inter-day precision and accuracy of the HPLC assay for ceftaroline in plasma.

	Nominal Concentration (µg/mL)	Concentration Measured (µg/mL) mean ± SD	Precision CV (%)	RE (%)
Intra-day (*n* = 5)	0.25 (LLOQ)	0.26 ± 0.01	2.62	2.96
	0.75	0.76 ± 0.02	2.05	1.39
	15	15.20 ± 0.54	3.56	1.35
	30	28.94 ± 0.34	1.17	3.52
Inter-day (*n* = 15)	0.25 (LLOQ)	0.26 ± 0.01	3.77	4.08
	0.75	0.77 ± 0.04	5.11	3.03
	15	14.64 ± 0.78	5.30	2.37
	30	28.06 ± 0.91	3.23	6.47

CV: coefficient of variation; LLOQ: lower limit of quantification; RE: relative error; SD: standard deviation.

**Table 5 pharmaceutics-13-00959-t005:** Stability of ceftaroline in plasma under different conditions.

Ceftaroline Concentration	Low QC 0.75 µg/mL (*n* = 3)			Medium QC 15 µg/mL (*n* = 3)			High QC 30 µg/mL (*n* = 3)		
	Mean ± SD	CV (%)	RE (%)	Mean ± SD	CV (%)	RE (%)	Mean ± SD	CV (%)	RE (%)
−20 °C (2 weeks)	0.68 ± 0.02	2.95	9.33	13.01 ± 0.17	1.29	13.28	25.85 ± 0.12	0.47	13.85
−80 °C (3 months)	0.65 ± 0.01	1.13	13.38	13.90 ± 0.16	1.12	7.35	27.36 ± 0.35	1.28	8.79
Three freeze/thaw cycles	0.68 ± 0.03	4.68	9.51	13.50 ± 0.15	1.08	10.03	26.68 ± 1.16	4.34	11.07
Sample processing (2 h)	0.66 ± 0.02	2.48	12.47	15.65 ± 0.02	0.15	4.30	32.30 ± 0.48	1.49	7.65
Autosampler 4 °C	8 h								

CV: coefficient of variation; QC: quality control; RE: relative error; SD: standard deviation.

**Table 6 pharmaceutics-13-00959-t006:** Pharmacokinetic parameters of ceftaroline in the two critically ill patients receiving 600 mg every 8 h.

Patient	Dose (mg Every 8 h)	C_max_ (µg/mL)	C_min_ (µg/mL)	CL_T_ (L/h)	t_1/2_ (h)	Vss (L)	AUC_τ_ (mg h/L)
1	600	18.22	2.34	8.84	3.33	36.40	67.85
2	600	30.28	2.87	7.33	2.73	23.61	81.82

CL_T_: total body clearance; t_1/2_: elimination half-life; Vss: volume of distribution at steady-state; AUC_τ_: area under the concentration–time curve in a dosing interval at steady-state.

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
