# Peer review of "Quantification of Ceftaroline in Human Plasma Using High-Performance Liquid Chromatography with Ultraviolet Detection: Application to Pharmacokinetic Studies"

_pharmaceutics, 2021, doi:10.3390/pharmaceutics13070959_

Round 1
Reviewer 1 Report
The authors present the development and validation of an HPLC-UV method for the quantification of ceftaroline in human plasma. I would like to congratulate them for their manuscript because the experimental procedure is described sufficiently and the method seems applicable for pharmacokinetic studies. The authors follow the FDA/EMA guidelines for bioanalytical method validation and their data seem to be in line with method validation requirements. The work is acceptable for publication. Some minor comments for their manuscript. I would suggest to reference the FDA and EMA guidelines (line 63). Maybe use color for figure 1 and the representative chromatograms to distinguish which is which (blank or lloq? patient receiving or not ceftaroline?) I would suggest to use different chromatograms. Please also fix the figure numbering (2 figures 1). Considering the LC-MS/MS method, is the LLOQ comparable? This is also minor since as they state the LLLOQ is adequate but have to be mentioned as only these 2 methods are availableAuthor Response
Thank you very much for your comments. We have adapted the manuscript according to them.
I would suggest to reference the FDA and EMA guidelines (line 63).
We have added the FDA and EMA references according to the suggestion of the reviewer (now line 72).
Maybe use color for figure 1 and the representative chromatograms to distinguish which is which (blank or lloq? patient receiving or not ceftaroline?) I would suggest to use different chromatograms.
We agree with the reviewer's opinion and we have modified sections A and B of now Figure 2, adding colours to identify the chromatograms and changing the scale.
Please also fix the figure numbering (2 figures 1).
We thank the reviewer for his observation and we have corrected the numbering of the figures. Since we have added a new figure in the introduction (suggestion of reviewer 3), now the figure numbering has changed.
Considering the LC-MS/MS method, is the LLOQ comparable? This is also minor since as they state the LLLOQ is adequate but have to be mentioned as only these 2 methods are available.
We really thank the reviewer for the proposal. We have modified the paragraph about the LLOQ, including a comparison with LC-MS/MS method reported in literature (lines 265-266). The new paragraph is as follows:
In spite of the use of ultraviolet detection, the LLOQ established in our method (0.25 µg/mL) is comparable to that reported by other authors applying a LC-MS/MS method (0.2 µg/mL)
Reviewer 2 Report
Manuscript Number: Pharmaceutics-1266274
Title: Quantification of ceftaroline in human plasma using high-performance liquid chromatography with ultraviolet detection: application to pharmacokinetic studies
Comments to Authors
Recommendation: Minor revision
The article " Quantification of ceftaroline in human plasma using high-performance liquid chromatography with ultraviolet detection: application to pharmacokinetic studies” aimed to develop a rapid, simple and reproducible method for the quantification of ceftaroline in plasma samples by HPLC-UV.
Although the presented work is well written and well designed, the used methodology appropriate, prior to publishing, several adjustments and corrections should be made:
Abstract:
Please, avoid abbreviations in the abstract (PK/PD)
Introduction:
Prior to abbreviation used, introduce the full term (page 1, line 33).
Page 1, line 35, correct “gran”
Please, write Gram-……, instead of “gram-… (please, check and correct throughout the text)
Page 2, line 45, line 51: please, uniform the writing - use either β-lactam, or beta-lactam
Materials and Methods
Page 2, line 77: please, be specific what “critically ill patients” mean, and add the explanation into text (do their therapy comprise not only ceftaroline, but another antibiotic, as well, or different medicines….)
Page 3, line 124: What does “QC” samples mean?
Page 4, line 147: What does PK mean? (please, check throughout the text that first the full name was used, with introduced appropriate abbreviation)
Table 1: please, define APACHE, correct “album”….
Results:
Page 5, Figure 1: Please, explain the width of the ceftaroline peak in the graph C,D of 1 minute – it is unexpectedly broad, in comparison to its peak in A and B
Tables 3 and 4, please add the legend with explanation what abbreviations mean
Author Response
Thank you very much for your comments. We have adapted the manuscript according to them.
Abstract:
Please, avoid abbreviations in the abstract (PK/PD)
According to the reviewer, we have included the full term in the abstract.
Introduction:
Prior to abbreviation used, introduce the full term (page 1, line 33).
We have changed the text so that the full term appears before the abbreviations.
Page 1, line 35, correct “gran”
We are thankful for the reviewer’s observation and we are sorry for the mistake. In fact, that was a typing error that we have corrected.
Please, write Gram-……, instead of “gram-… (please, check and correct throughout the text)
We have corrected the spelling mistake throughout the text.
Page 2, line 45, line 51: please, uniform the writing - use either β-lactam, or beta-lactam
We have made it uniform, using always beta-lactam.
Materials and Methods
Page 2, line 77: please, be specific what “critically ill patients” mean, and add the explanation into text (do their therapy comprise not only ceftaroline, but another antibiotic, as well, or different medicines….)
We have added an explanation of critically ill patients (lines 74-77).
Page 3, line 124: What does “QC” samples mean?
We have explained the meaning of QC the first time it appears (line 86), and used the abbreviation along the manuscript.
Page 4, line 147: What does PK mean? (please, check throughout the text that first the full name was used, with introduced appropriate abbreviation)
The full name of pharmacokinetics and the abbreviated form (PK) are now in the last paragraph of the introduction section (line 75).
Table 1: please, define APACHE, correct “album”….
We have corrected the error (albumin) and we have defined the meaning of APACHE in the table 2 legend.
Results:
Page 5, Figure 1: Please, explain the width of the ceftaroline peak in the graph C,D of 1 minute – it is unexpectedly broad, in comparison to its peak in A and B
The reason of the difference in the peak width is due to the great difference in concentration (LLOQ: 0.25 µg/mL, standard: 20 µg/mL). When all the chromatograms of the standard curve samples are overlapped, this effect is very clear.
Tables 3 and 4, please add the legend with explanation what abbreviations mean
We have added a legend in both tables with the explanation of abbreviations.
Reviewer 3 Report
Ceftaroline is a fifth-generation cephalosporin that has its role in anti-infective therapy. However, there are currently few methods of analysis for fifth-generation cephalosporins. Therefore, I appreciate the development of a simple, rapid, and reproducible analysis method for this compound.
The article needs some improvement in terms of content and writing.
1.Introduction. To better value the manuscript, it is necessary to introduce more detailed information regarding the physicochemical properties of the targeted substance. In this regard, I suggest a table (solubility in essential solvents, pKa, log P, etc.) These data are critical to the process of optimizing and validating the HPLC method. In addition, a scheme that suggests ceftaroline metabolism is helpful for general understanding. For example, is there a possibility that the inactive metabolite will interfere with the determinations?
2.Materials and Methods. I did not find in the manuscript the optimization of the HPLC method (preliminary steps). Was the technique successful by chance? I refer to the conditions and parameters used (selection of the column, mobile phase, etc.). The wavelength of 238 nm was selected based on literature or after the experimental registred UV spectra?
3.Results. The number of ill patients is minimal. The concentration-time profile of ceftaroline in the two patients is similar. However, for many patients, the situation could be very different, depending on individual variability. It is necessary to mention the weak points of the study and the stages in which various errors are possible.
Line 39, page 1. Remove „US” when you use the abbreviation FDA.
Line 42, page 1. Correct „pediataric” word.
Line 124, page 3. Please, define QC samples in the manuscript.
Line 148, page 4. „q8h” is unusual and confusing all over the manuscript.
Table 1. Correct „Album” in Table 1. Define APACHE in the food note of the table.
Author Response
Thank you very much for your comments. We have adapted the manuscript according to them.
Introduction. To better value the manuscript, it is necessary to introduce more detailed information regarding the physicochemical properties of the targeted substance. In this regard, I suggest a table (solubility in essential solvents, pKa, log P, etc.) These data are critical to the process of optimizing and validating the HPLC method. In addition, a scheme that suggests ceftaroline metabolism is helpful for general understanding. For example, is there a possibility that the inactive metabolite will interfere with the determinations?
We have implemented the introduction of the manuscript with information about the physicochemical properties of ceftaroline (table 1 in the new version) and with its metabolic profile (figure 1 in the new version).
Concerning metabolism, the following sentence has been added (lines 262-264):
Ceftaroline is eliminated mainly by renal route, and only small fraction is converted into an inactive metabolite, which presents higher polarity and therefore, interference with ceftaroline is not expected.
- Materials and Methods. I did not find in the manuscript the optimization of the HPLC method (preliminary steps). Was the technique successful by chance? I refer to the conditions and parameters used (selection of the column, mobile phase, etc.). The wavelength of 238 nm was selected based on literature or after the experimental registred UV spectra?
We appreciate this comment very much. We have added the following paragraph in material and method section (lines 98-101):
Final conditions of the method were selected from preliminary experiments, in which different columns, mobile phase composition, flow rate, and sample preparation conditions were tested. An UV spectrum was performed in order to select the wavelength for detection.
- Results. The number of ill patients is minimal. The concentration-time profile of ceftaroline in the two patients is similar. However, for many patients, the situation could be very different, depending on individual variability. It is necessary to mention the weak points of the study and the stages in which various errors are possible.
The reviewer is right, and only two patients is very low to obtain conclusions about the PK of ceftaroline. However, the purpose of the study was to develop and validate the method to quantity the antibiotic. The concentration-time profile in the two patients we present in the manuscript was to demonstrate the application of the method for PK studies. It is well known that, due to the high variability of the PK behaviour, a high number of patients must be included in this kind of studies, and population PK model should be used.
We have included a paragraph in the discussion (lines 329-333):
It is well known that the variability of PK of drugs in critically ill patients is very high [33], and clinical trials to estimate the PK parameters in this population must include a high number of patients; additionally, population models are recommended. In this regard, our study presents a limitation. However, our purpose was to demonstrate the usefulness of the method better than a full characterization of the ceftaroline PK in critically ill patients.
Line 39, page 1. Remove „US” when you use the abbreviation FDA.
We have removed US.
Line 42, page 1. Correct „pediataric” word.
We are thankful for the reviewer’s observation and we are sorry for the mistake. In fact, that was a typing error that we have corrected.
Line 124, page 3. Please, define QC samples in the manuscript.
We have explained the meaning of QC the first time it appears (line 86), and used the abbreviation along the manuscript.
Line 148, page 4. „q8h” is unusual and confusing all over the manuscript.
We have replaced ``q8h´´ by ``every 8h´´ all over the manuscript.
Table 1. Correct „Album” in Table 1. Define APACHE in the food note of the table.
We have corrected the error (albumin) and we have defined the meaning of APACHE in the table 2 legend.